# Accurate Quantum States for a 2D-Dipole

Daniel Vrinceanu 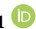

Department of Physics, Texas Southern University, Houston, TX 77004, USA; daniel.vrinceanu@tsu.edu

**Abstract:** Edge dislocations are crucial in understanding both mechanical and electrical transport in solid and are modeled as line distributions of dipole moments. The calculation of the electronic spectrum for the two dimensional dipole, represented by the potential energy $V(r, \theta) = p \cos \theta / r$, has been the topic of several studies that show significant difficulties in obtaining accurate results. In this work, we demonstrate that the source of these difficulties is a logarithmic contribution to the behavior of the wave function at the origin that was neglected by previous authors. By taking into account this non-analytic deviation of the solution of Schrödinger's equation, superior results, with the expected rate of convergence, are obtained. This goal is accomplished by "adapting" general algorithms for solving partial derivative differential equations to include the desired asymptotic behavior. We illustrate this principle for the variational principle and finite difference methods. Accurate energies and wave functions are obtained not only for the ground state but also for the first eleven excited states and are useful for designing nanoelectronic devices. This paper demonstrates that augmentary knowledge about analytic properties of the solutions leads to the improved convergence and stability of numerical methods.

**Keywords:** edge dislocations; electronic states; numerical methods

## 1. Introduction

Imperfections within a crystal structure that otherwise is a perfectly ordered solid-state arrangement of atoms and molecules are responsible for a number of useful attributes, including the alteration of conductivity or optical properties. Such imperfections take the form of vacancies, where atoms are missing from their lattice positions; interstitials, where atoms sit in small gaps in between lattice positions; or defects, where atoms different than all of the others sit at the lattice position.

An edge dislocation in a solid is a type of crystallographic imperfection that occurs when an extra half crystal plane of atoms is inserted into the crystal lattice. The edge dislocation is characterized by a line defect, known as the dislocation line, where the lattice is locally distorted. This can be visualized as being caused by the termination of a plane of atoms in the middle of the crystal. In such a case, the surrounding planes of atoms are not straight but instead bend around the edge of the terminating plane so that the crystal structure is perfectly ordered on either side.

The theory of elasticity of materials with defects was originally developed by Vito Volterra in 1907 [1]. In 1934, G. I. Taylor, among others, proposed the edge dislocations form the explanation for the low stresses observed to produce plastic deformation compared to the much greater theoretical predictions [2]. The mechanical properties of ductile materials are well explained by edge dislocation as carriers of plastic deformation because the energy to move them is much less than the energy to break entire plane of atoms at once [3]. When dislocations interact with grain boundaries and surfaces, several strengthening mechanisms are observed, for example, the Hall–Petch effect [4]. Another example is the enhancement of the yield strength in surface dominated nanostructures [5].

The transport properties and binding of impurities associated with edge dislocations relate to the superconducting properties of the solid [6]. In a supersolid state, atoms and

molecules are arranged in a regular crystal lattice structure, characteristic of a solid, but they also flow without any viscosity or resistance due to quantum tunneling mediated by defects and vacancies in the crystal lattice. These quantum defects lead to a peculiar form of coherence in the arrangement of particles in a solid, allowing for the simultaneous existence of crystalline order and superfluidity. It was demonstrated [6] that edge and screw dislocations, which are manifestations of disorder, actually contribute to the coherence required for superfluidity. This phenomenon is akin to the Mott insulator transition to superfluidity observed in quantum gases confined to optical lattices as a realization of the Bose–Hubbard model [7].

The electronic states and structure of edge dislocations determine the macroscopic mechanical and electric properties of materials and have been been modeled with ab initio methods, for example, [8]. However, the accuracy of these Density Functional Theory quantum calculations is limited to few hundred atoms in the computational primitive unit cell. Effective models have been used to evaluate the increase in the drain current of metal oxide semiconductor field effect transistors (MOSFETs) due to the presence of edge dislocations in the channel [9]. In this case, the dislocation is modeled effectively as a nanowire. Since the geometry of atoms in edge dislocation is not compatible with periodic boundary conditions, artificial dipole and quadrupole configurations have been proposed.

The precise knowledge of the electronic quantum states for an edge dislocation is important for evaluating the potential use of edge dislocation in quantum computing. Specific physical systems are designed for use as qubits because the basic units of quantum information require several critical properties such as low decoherence and optical addressability.Low dimensional solid state defects were proposed as candidates for quantum processing. For example, nitrogen vacancies in diamond are very robust and easy to initialize, manipulate, and measure with high fidelity at room temperature. Moreover, their ability to couple with their environment makes them excellent quantum sensors [10]. The key feature for the success of this defect for quantum sensors is the highly localized electronic bound states confined to a region on the scale of a single lattice constant. This makes the defects very bright and easily coupled with optical lasers. These kinds of defects are called Faber centers, color centers, or chromophores because electrons can undergo optical transitions between the orbitals of the defect by emitting photons at specific wavelengths. Quantum dots are also characterized by tight electron confinement, covering an area of several, up to hundreds, of lattice constants, but still smaller than the wavelength of the emitted radiation [11]. Many other zero dimensional candidates were proposed and systematically investigated [12] as q-bit candidates. The electron quantum states of one-dimensional defects, such as edge dislocations, may also play roles in quantum processing [13], combining the localization of the electron state in a place perpendicular to the direction of the defect with free electron motion along the edge.

Vacancies in the crystal lattice tend to be occupied by conduction electrons that migrate from the last atoms that form the edge of the dislocation, which, in this way, develops a local electric dipole moment pointing towards the vacancy. This dipole moment is linearly and uniformly distributed along the dislocation line and may confine the electrons in the edge proximity. The goal of this paper is to clarify the difficulties related to the calculation of the localized electronic states of the edge dislocations and provide accurate energy levels and wave functions. The remainder of this paper is organized as follows. Section 2 describes the effective model for the electronic Hamiltonian for an edge dislocation, and the logarithmic singularity at the origin. Section 3 discusses the nature of the spectrum of quantum states for the edge dislocation, the labeling of energy levels, and the features of the wave functions corresponding to each quantum level. Section 4 explains how the knowledge of the behavior of the wave function around the singularity, which is not regular as in the case of analogous three dimensional singular potentials, helps improve the convergence of numerical methods by directly embedding the logarithmic singularity directly into the numerical scheme. Note that conventional numerical methods for solving Schrödinger partial derivatives equations, such as Finite Difference or the Rayleigh–Ritz

variational principle, assume that solutions have regular analyticity properties and have slow convergence for problems that are known to have non-analytic solutions, such as the 2D dipole electronic states considered here.

## 2. Electric Dipole in Two Dimensions

If the straight edge of the dislocation is oriented along the $z$ axis, then the electronic potential energy is well described by the potential of a uniform linear distribution along the $z$ axis of dipoles pointing in the $x$ axis direction [14]:

$$V(r,\theta) = \frac{e\delta}{2\pi\epsilon_0}\cos\theta/r = p\cos\theta/r \qquad (1)$$

This potential energy is given in cylindrical coordinates $(r, \theta, z)$, where $\delta$ is the linear dipole density, $e$ is the elementary charge, and $\epsilon_0$ is the vacuum permittivity. The edge dipole density $\delta$ depends on the details of the specific solid, such as Fermi energy and lattice constant, and has the dimension of a charge (electric dipole moment per unit length). For example, for an ionic fcc crystal, such as rock salt, $\delta$ is $0.5e$ because the basis for the crystal has a $Cl^-$ ion at $(0, 0, 0)$ and a $Na^+$ ion at $(0.5, 0, 0)$. This potential is $z$-invariant; therefore, this problem is essentially two dimensional in the $(x - y)$ plane. On using $p$-dependent length $\hbar^2/(2mp)$ and energy $2mp^2/\hbar^2$ units, the time-independent Schrödinger equation is written as a dimensionless eigenvalue problem

$$-\nabla^2\psi + \frac{\cos\theta}{r}\psi = \epsilon\psi \qquad (2)$$

where the energy of the eigenstate is $\epsilon$ in $p$-dependent units. The effective mass $m$ of the electron in the solid is typically a small fraction of the electron mass, about $\sim 0.2\ m_e$. We see that with these values, the unit of length for this problem is approximately 0.3 nm, and the unit of energy is approximately 2.72 eV.

The two dimensional quantum dipole Equation (2) has two apparent difficulties: it is not separable and it is singular at the origin, with a Coulomb type singularity. Getting the energy of the ground state has been attempted by using several methods, as revealed in some details in a paper by Dasbiwas et al. [14], with non-converging or slowly converging results. The quest for finding the ground state energy of the quantum 2D dipole was started by the pioneering work of Landauer in 1954 using a variational approach [15]. A semiclassical trajectory study [14] has confirmed the general features for the density of energy for this system and the chaotic dynamics characteristic space-filling orbits and sensitivity on initial conditions. It was noted that some classically allowed regions were not visited by the orbits, indicating a possible lack of ergodicity, or some hidden symmetries of the system.

It is surprising, giving the manifold increase in computational power available, that this recent paper [14] reports only a modest improvement in accuracy of the ground state at the level of 1%. More recent work [16–18] has reported significant progress in obtaining more accurate values not only for the ground state but also for excited states. However, these new publications, as well as all other previous attempts, acknowledge unexpected difficulties, such as a slow convergence rate and numerical instabilities.

The goal of this paper is to explain that the source of difficulties in obtaining accurate results for the spectrum of the quantum dipole is *the logarithmic behavior of the wave function at the origin*, a feature that has not been recognized in any previous reports. Indeed, the results presented here are obtained with numerical methods that directly and explicitly take into account the non-analyticity of the wave function and that acquire the expected rates of convergence in a stable fashion by respecting the peculiar behavior of the wave function at the origin.

The underlying assumption in many approximation methods used to solve Equation (2) is that the wave function is analytic everywhere, including at the origin. This is true in general for the Finite Difference Method (FDM) and explains the poor performance in solving (2). However, as is shown below, FDM can be modified so that the correct

asymptotic form is explicitly assumed. The results using the modified, asymptotically adapted FDM show a dramatic improvement of convergence rate. Results proving the same point are also obtained using a variational Rayleigh–Ritz method. The correct asymptotic form at the origin is simply included by extending the variation basis set of trial functions with logarithmic terms.

It was observed in [14] that the 2D dipole potential can be realized by bringing together two infinite linear distributions of charges with the same density but opposite signs. This suggests the 2D hydrogen atom wave functions as a reasonable variational basis set. Amore and Fernandez [17] take this point further, hypothesizing that the convergence problems are generated by the lack of inclusion of continuum wave functions. Therefore, they propose the use of a Slater-type 2D hydrogenic basis set. However, the two-dimensional hydrogen atom has actually two realizations. One realization has a point-like positive charge center, and the wave function is constrained by some means to a plane. This is the basis set used in [14,17]. All wave functions in this basis set are analytical at the origin, and therefore are not suitable to describe non-analytic wave functions solutions of (2). The second realization of the 2D hydrogen atom is for a line distribution of positive charge. The Coulomb potential in this case is logarithmic and the electronic states in such potential also have logarithmic terms in their expansion about the origin, as demonstrated by Gesztesy and Pittner [19]. It is therefore natural to expect some logarithmic contributions to the wave functions of electrons in the field of two lines of charges of opposite sign, even in the limit of zero distance between them when they form the 2D dipole potential of an edge dislocation. Therefore wave functions for this kind of 2D hydrogen atom are more suitable to form a correct basis set for solving the 2D dipole problem (2). Unfortunately, no simple close form solution are known for the 2D hydrogen problem.

## 3. Quantum States for the 2D Dipole

The quantum 2D dipole problem (2) is truly two dimensional because no separation of variable is possible. However the potential has a reflection symmetry with respect to the $x$-axis, and therefore there are solutions of (2) with definite parity with respect to $y \rightarrow -y$ transformation. Odd states are zero along the $x$-axis, while the even states have zero $y$-derivative along the $x$-axis.

Figure 1 shows the energy spectrum for the two sets of states. The ground and the first even excited states lie under the first odd state. The 4th and the 5th odd states are almost (accidentally) degenerate. Each state is also labeled by two quantum numbers $n$ and $m$ which are loosely associated with the principle quantum number and angular momentum quantum number, although due to the lack of cylindrical symmetry the angular momentum is not conserved. The meaning of these quantum numbers is clear from Figures 2 and 3 that show density plots for the even and, respectively, odd states. These states correspond to the states represented in Figure 1. The quantum number $n$ is equal to the number of "vertical" nodal lines, while the quantum number $m$ is given by the number of "horizontal" nodal lines. Since the problem is not exactly separable, the nodal lines can not be exactly vertical or horizontal. However, these two kinds of nodal line are clearly distinguishable, at least for lowest states shown in these figures.

Despite the Coulomb $1/r$ singularity, the 3D and 2D restricted hydrogen wave functions are analytic at the origin and have solutions that involve familiar Laguerre functions [20]. The 2D name here emphasizes that the motion of the electron around the point-like positive charge is constrained in a plane. However, when the electron moves around a line of positive charge the wave function is not analytic and has a logarithmic term when expanded around the origin. This suggests that this is also true for the case of a 2D dipole potential (1). Indeed, the local expansion for the wave function is

$$\psi(r, \theta) \approx 1 + \frac{x}{4} \log(r) + \cdots, \qquad (3)$$

which shows that although the wave function is finite at the origin, it is *not analytic*, having logarithmic terms. This is the source of difficulties for conventional numerical methods that assume that solutions are analytical at all points.

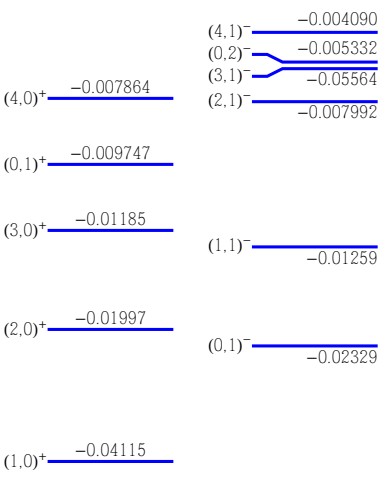

**Figure 1.** The first six even (left panel) and six odd (right panel) energy levels labeled as $(n, m)^\pi$, where $n$ and $m$ are quantum numbers and $\pi = \pm$ is the parity of the state.

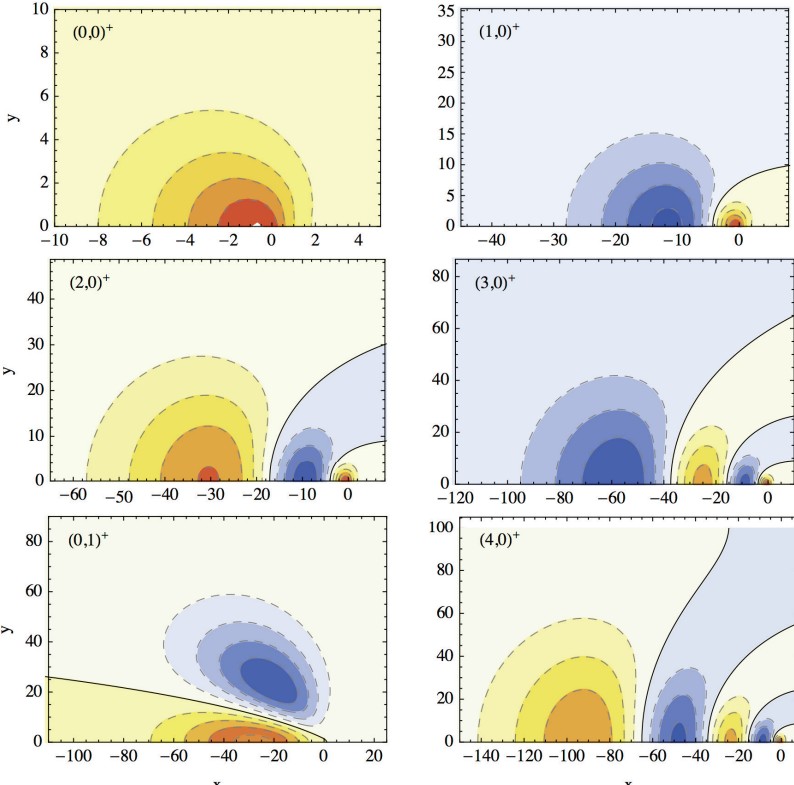

**Figure 2.** Contour levels (dashed lines) for wave functions for the first six even eigenstates for Equation (2) labeled as $(n, m)^+$. Solid lines show nodal lines. Yellow/orange/red colors represent regions were the value of the wave function is positive in increasing order from yellow to red, while pale blue/blue represent negative values.

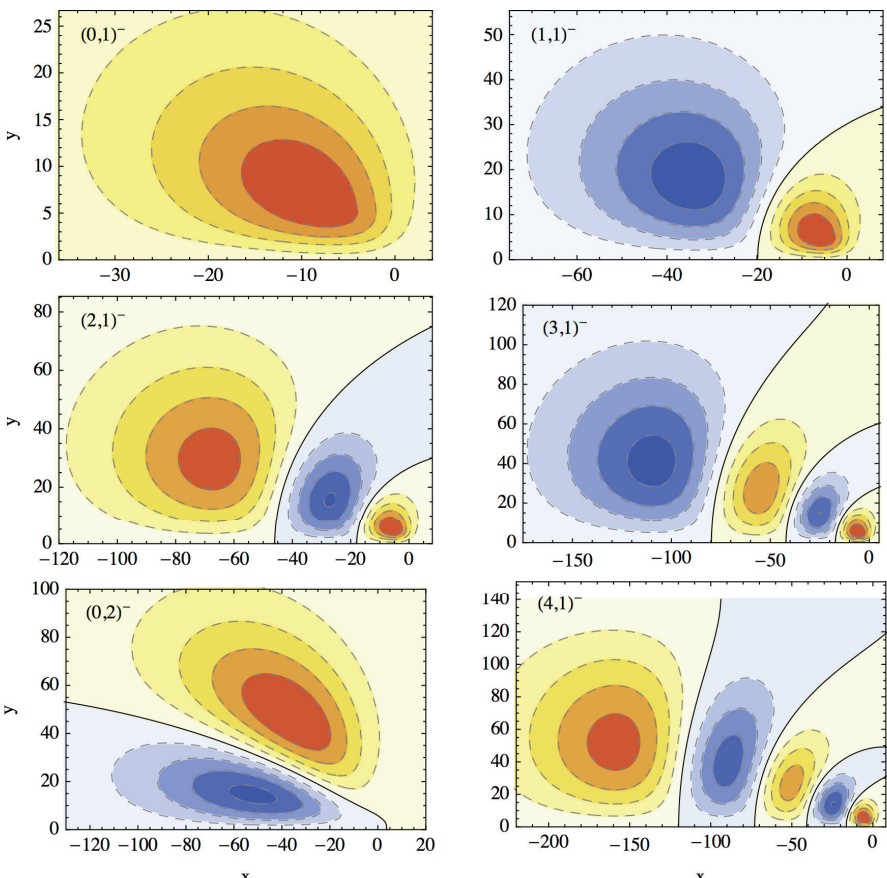

**Figure 3.** Contour levels (dashed lines) for wave functions for the first six odd eigenstates for Equation (2) labeled as $(n, m)^-$. Solid lines show nodal lines. Yellow/orange/red colors represent regions were the value of the wave function is positive in increasing order from yellow to red, while pale blue/blue represent negative values.

## 4. Finite Difference and Variational ABC

The majority of computational methods used for solving Schrödinger's equation rely on certain regularities of the wave function, including analyticity. For example, the familiar Finite Difference five-point stencil provides the correct discretization of the Laplacian on a grid of size $h$ as

$$(\nabla^2 \Psi)_{i,j} = (\Psi_{i+1,j} + \Psi_{i-1,j} + \Psi_{i,j+1} + \Psi_{i,j-1} - 4\Psi_{i,j})/h^2$$

only for an analytical wave function. It is therefore not surprising that standard methods fail to converge, or converge very slowly, for non-analytical wave functions, even for removable singularities. This phenomenon, and a straight forward way to address this problem, has been studied in [21], where the Finite Difference method was shown to work correctly providing it is modified to include the Asymptotic Behavior Correspondence (ABC). This involves using a method that is able to inherit the asymptotic behavior of the wave function near the singularity from the continuous Schrdinger equation to the discretized one. This principle of embedding a known local feature of the wave function into the numerical scheme is more general than Finite Difference. It applies, for example, to Variational Methods, where by including the basis function that has the correct asymptotic behavior, one can improve the convergence of the method, which might be very slow, or even converge to the wrong result if the trial functions in the basis set do not have correct asymptotic behavior.

For a regular rectangular Cartesian grid of the form: $x_i = ih$ and $y_j = jh$, with $i$ and $j$ integer numbers, the direct discretization of the potential energy (1)

$$V_{i,j} = V(x_i, y_j) = \frac{1}{h} \frac{i}{i^2 + j^2}$$

clearly shows the difficulty for the central grid point $(0,0)$. The ABC prescription to regularize the potential is to replace $V_{ij}$ with a discrete potential that is finite at the origin of the following form:

$$V_{ij} = \frac{1}{h} q_{ij}$$

where $q_{ij}$ correction is obtained by making sure that $\Phi = 1 + x/4 \log(x^2 + y^2)$ is an asymptotic solution for the discretized equation in the $h \to 0$ limit. This condition leads to

$$
\begin{aligned}
q_{ij} = & -i \log(i^2 + j^2) + (i+1)\log((i+1)^2 + j^2)/4 + \\
& + (i-1)\log((i-1)^2 + j^2)/4 + i\log(i^2 + (j+1)^2)/4 + i\log(i^2 + (j+1)^2)/4
\end{aligned}
$$

which is finite at all grid points and approaches $i/(i^2 + j^2)$ for large indices $i$ and $j$.

The Finite Difference five-point ABC stencil was implemented using the PETSc parallel library [22], and the first six eigenvalues and eigenfunctions were obtained using the associated SLEPc library [23]. A variant with a nine-point stencil was also implemented, but the results did not reveal significant differences in term of accuracy or speed. Taking advantage of symmetry, the calculations for odd and even states can be made separately by changing the boundary conditions along the $x$-axis, where odd states are zero and even states have vanishing derivatives along the $y$-axis. On all other borders of the domain, the wave function was set to zero. The calculation took advantage of the parallelism of SLEPc via the MPI protocol to distribute the calculation to a computer cluster of 32 compute nodes, with a total of 512 processing cores.

The computational domain was discretized with a rectangular grid with an increasing number of nodes $N$ along the $y$ axis, from 100 to 3200. The size of the mesh, indicating the size of the Hamiltonian square matrix that needs to be diagonalized, increases proportional to $N^2$ with a maximum size of up to 2.15 million. However, due the local nature of the Laplacean operator, the Hamiltonian matrix is sparse, matrix vector operations are quite fast, and the actual matrix does not have to be stored in memory. The C++ PETSc/SLEPc library used in calculation operates on sparse matrices and is designed to work on parallel computers.

Table 1 shows that the expected quadratic convergence rate for both odd and even states is obtained as the number of nodes $N$ increases. The row marked as $\infty$ represents the limit results obtained using the Shanks re-summation technique [24]. Shanks transformation is a numerical technique used to accelerate the convergence of a series; it is named after the American mathematician Daniel Shanks. The transformation involves forming pairs of adjacent partial sums of the series and using these pairs to estimate the limit of the series. The transformed sequence is expected to converge more rapidly to the true limit of the series, providing a more accurate approximation.

The last row in Table 1, marked RRM, represents the results obtained using the Rayleigh–Ritz variational method. In this method, we find the eigenvalues and eigenfunctions for Equation (2) by searching to find the minimum of a functional within a space of trial functions spanned by a basis set of functions that are expected to capture the essential features of the true solution. The functional is the expectation value of the Hamiltonian. We choose a basis set that contains both the analytical 2D hydrogen Sturmian/Slater-type functions $\{r^j e^{-\alpha r} \cos n\theta\}_{j,n}$ as well as logarithmic terms of the form $\{\log(r)^p r^j e^{-\alpha r} \cos n\theta\}_{p,j,n}$, with $j, n, p = 0, 1, 2, \ldots$. For each eigenvalue, the optimal $\alpha$ coefficient and the mixing coefficients are calculated such that the functional is minimized. Excited states are constructed to be orthogonal on lower states. Excellent results are obtained, with at least six digits of

precision for energy, by using no more than 108 terms in the augmented basis set. In the absence of logarithmic terms in the basis set, the convergence is very poor.

**Table 1.** Convergence of ABC Finite element method with increasing number $N$ of discretization nodes along the $y$-axis, for the first six even and odd quantum states of the 2D dipole labeled as $E_1$ to $E_6$. The line marked as $\infty$ indicates the results obtained using a Shanks transform to accelerate the convergence of the sequence of approximations. The line marked as RRM shows the results of Rayleigh–Ritz variational method that includes non-analytic terms in the basis set of trial functions.

| N | $E_1$ | $E_2$ | $E_3$ | $E_4$ | $E_5$ | $E_6$ |
|---|---|---|---|---|---|---|
| | | | Even states | | | |
| 100 | −0.132293 | −0.0401750 | −0.0196393 | −0.0117061 | −0.00974627 | −0.00779385 |
| 200 | −0.135950 | −0.0408653 | −0.0198759 | −0.0118145 | −0.00974603 | −0.00785253 |
| 400 | −0.137231 | −0.0410777 | −0.0199470 | −0.0118468 | −0.00974613 | −0.00786994 |
| 800 | −0.137610 | −0.0411375 | −0.0199668 | −0.0118558 | −0.00974609 | −0.00787475 |
| 1600 | −0.137713 | −0.0411533 | −0.0199721 | −0.0118581 | −0.00974611 | −0.00787603 |
| 3200 | −0.137739 | −0.0411574 | −0.0199734 | −0.0118587 | −0.00974610 | −0.00787648 |
| ∞ | −0.137748 | −0.0411588 | −0.0199739 | −0.0118590 | −0.00974610 | −0.00787656 |
| RRM | −0.137748 | −0.0411588 | −0.0199738 | −0.0118587 | −0.00974608 | −0.00787432 |
| | | | Odd states | | | |
| 100 | −0.0227367 | −0.0123917 | −0.00789998 | −0.00551321 | −0.00529711 | −0.00408227 |
| 200 | −0.0229585 | −0.0124666 | −0.0079347 | −0.00553231 | −0.0053128 | −0.00409397 |
| 400 | −0.0231171 | −0.012523 | −0.00796158 | −0.00554735 | −0.00532098 | −0.00410329 |
| 800 | −0.0232037 | −0.0125541 | −0.00797645 | −0.0055557 | −0.00532511 | −0.00410846 |
| 1600 | −0.0232482 | −0.0125701 | −0.00798409 | −0.00555999 | −0.00532719 | −0.00411113 |
| 3200 | −0.0232706 | −0.0125781 | −0.00798795 | −0.00556215 | −0.00532823 | −0.00411247 |
| ∞ | −0.0232932 | −0.0125862 | −0.00799186 | −0.00556432 | −0.00532937 | −0.00411379 |
| RRM | −0.0232932 | −0.0125863 | −0.00799184 | −0.00556419 | −0.00533114 | −0.00411271 |

It is clear that the dramatic improvements in both the Finite Element and Variational method derive from the explicit inclusion of logarithmic terms within the respective numerical scheme. Numerical schemes that ignore the troublesome behavior of the wave function at the origin have poor performance.

## 5. Conclusions

Accurate energy and wave functions were obtained for the ground quantum state and several excited states of the 2D dipole. The two dimensional linear dipole excellently models the potential energy seen by electrons in edge dislocations in crystalline solids. The electronic states of these defects have a significant effect on the transport, elastic, and superconducting properties of the solid. The logarithmic behavior of the wave function for the 2D dipole is confirmed close to the origin. It is demonstrated that computational methods that embed this behavior and adapt to include the expected asymptotic limit have an expected convergence rate and do not falter as in previously reported works that failed to recognize the peculiar features of the wave function close to the origin. Both the finite difference method and the Rayleigh–Ritz variational principle have the anticipated numerical performance when the singularity is explicitly included in the numerical scheme. As expected, core penetrating low angular momentum s-wave-type wave functions are more affected by the logarithmic terms than the p and d-type state that avoids the origin. All of the calculated wave functions fill a wedge-shaped region instead of the classically allowed circular disk. Ongoing and future efforts aim at evaluating the interaction between neighboring edge dislocations, given the considerable spatial extent of the localized electronic quantum states that leads to a strong interaction. This work can be extended to 2D magnetic dipole systems that have recently been the focus of extensive studies [25].

**Funding:** This research was funded by National Science Foundation grant number PHY-1831977.

**Data Availability Statement:** The software used to obtained the data presented in this study as well as the data itself are available on request from the author.

**Acknowledgments:** This work has received support from the High Performance Computing Center at Texas Southern University.

**Conflicts of Interest:** The authors declare no conflicts of interest.

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
