# Peer review of "Accurate Quantum States for a 2D-Dipole"

_nanomaterials, doi:10.3390/nano14020206_

Round 1

Reviewer 1 Report

Comments and Suggestions for Authors

This paper deals with the study of quantum states of a 2D dipole. This problem is relevant to the understanding of solid states defects. The paper is sound and well written. it also contains a very interesting discussion on the nature of the dipole solutions near origine. For that reason, I think that it could be accepted for publication, after a minor revision. In particular, I would suggest the following small changes:

1 - The purpose of the paper is missing in the Introduction. A short description of the paper content should therefore be included.

2 - Ref [6] is  incomplete and should be corrected.

3 - I noticed several (too many) misprints. The author should carefully read the paper.

Author Response

All points highlighted in reviewer's report were addressed and the manuscript was revised accordingly:

  • Introduction has more explanations and motivations for this work and a description of organization of the manuscript was added
  • ref [6] was corrected
  • the manuscript was screened for typos and corrected 

Reviewer 2 Report

Comments and Suggestions for Authors

The authors explains a method to include the non-analytic contribution of a wave function for solving differential equations by finite differences. They illustrate their method by calculations of the wave function in a 2D dipole potential. 

I think the article is clear and provides results which may be of general use. Therefore I recommend publication.  However, figure and table captions should be enlarged to provide a more detailed description. In Figure 2 and 3, the color code should be given, table 1 should explain also the meaning of N, E_1,..., RRM.

Author Response

The point raised by the reviewer was address by expanding the caption for two figures and the table in the manuscript.